# A Hybrid Path-Planning Strategy for Mobile Robots with Limited Sensor Capabilities

**DOI:** 10.3390/s19051049

**Published:** 2019-03-01

**Authors:** Guilherme Carlos R. de Oliveira, Kevin B. de Carvalho, Alexandre S. Brandão

**Affiliations:** Núcleo de Especialização em Robótica—NERO, Departamento de Engenharia Elétrica—DEL, Universidade Federal de Viçosa—UFV, Viçosa MG 36570-900, Brazil; guilherme@ufv.br (G.C.R.d.O.); kevinbdc@gmail.com (K.B.d.C.)

**Keywords:** path planning, global and local planner, hybrid strategy, A* search, tangential escape

## Abstract

This paper introduces a strategy for the path planning problem for platforms with limited sensor and processing capabilities. The proposed algorithm does not require any prior information but assumes that a mapping algorithm is used. If enough information is available, a global path planner finds sub-optimal collision-free paths within the known map. For the real time obstacle avoidance task, a simple and cost-efficient local planner is used, making the algorithm a hybrid global and local planning solution. The strategy was tested in a real, cluttered environment experiment using the Pioneer P3-DX and the Xbox 360 kinect sensor, to validate and evaluate the algorithm efficiency.

## 1. Introduction

Mobile robotics is one of the major study fields within robotics. To achieve autonomous navigation, the agent must be able to plan a strategy that will efficiently, wihtin a feasible amount of time, lead to its destination point. An agent is truly autonomous, when it takes its own actions with no human interaction in order to avoid collision with obstacles in the environment. In most situations, we need to assume that obstacles may change location in the world, and that it will only be noticed when the robot moves around, in a so called dynamic environment.

To solve this planning problem, a family of algorithms based on the classical AI graph search have been proposed over the years, which can be traced back to the famous A* search [1], a heuristic driven graph search. At environments are usually modeled as directional graphs or grids, the algorithm succeeds at finding an optimal path (if it exists, and using a fair heuristic) from a starting point to a goal destination. Even though the heuristic function speeds up the search, it becomes impractical to recalculate a path when environment changes are detected. In a real situation, it occurs whenever a new obstacle is detected or a new path is available. To overcome this limitation, famous algorithms with different, faster re-planning tools were proposed, such as D* [2] and focused D [3], which uses incremental updates to the graph edges in real time to obtain an optimal path. The LPA* [4] algorithm stores previous graph searches in order to decrease the execution time of future searches. The D* Lite [5], considered the state-of-art algorithm for the path planning using graph search, implements the same strategy as D*, but is more computationally efficient. Given the current state of the heuristic search family and the well established replanning strategies, most research today focuses on variants of the state-of-the-art algorithm that speed up the search and lowers the memory consumption of the solutions. Recent works, containing improvements and comparisons between the most popular graph search approaches can be found at [6,7].

On the other hand, some strategies deal with the planning problem while the robot is moving (online), based on real time and local sensor measurements, which is also called reactive navigation. It is the case of the famous artificial potential fields method (APF) [8], which creates artificial forces of repulsion and attraction that drives the robot to its destination. The vector field histogram (VFH) [9] models the environment as a two-dimensional cartesian histogram grid, that is updated incrementally while the robot is moving. The modeled grid is then reduced to a one-dimensional polar histogram, which is used to select the safest path based on each polar density sector. The VFH* [10], combines the A* algorithm and VFH to select sectors which will lead to better paths through the histogram grid. Another local obstacle avoidance algorithm, the tangential escape [11,12] creates a virtual temporary goal that leads the robot in a path tangent to the obstacle, thus avoiding the collision. These algorithms, though simple and very efficient in terms of memory and time-processing, can overcome local minima and get an optimal (or sub-optimal) convergence. Much research has focused on these problems, as shown in [13,14,15].

The present work approaches the path planning problem considering low budget platforms, thus limited computational capabilities. As we wanted an algorithm capable of finding an optimal path when enough information about the environment is provided and real time obstacle avoidance (for an unknown or dynamic world), we ended up creating a hybrid solution that combines heuristic search (global planning) for the offline phase with a real-time obstacle avoidance algorithm (local planning) for the online phase. As offline we consider each moment before the robot start its movements towards a destination point. At this point the global planning algorithm, A*, is used to search an optimal path from the current position to the goal. As online phase we consider every moment when the robot is moving, when it needs to deal with the obstacle avoidance task. At this phase it used the path obtained from the A* algorithm to avoid collisions with known obstacles and the local planning algorithm, tangential escape, to perform obstacle avoidance for the unknown obstacles, returning to the path initially planned after overcoming the obstacles. In the literature, hybrid strategies, combining other popular global and local planning methods, have been presented in [10,16,17,18]

## 2. The Problem and Its Constraints

Given the path planning problem for a 2D environment, where a state describes a robot pose (world coordinates x,y and the robot orientation ψ), we must provide a set of states, from a starting *S* and a final goal *G* ones. We are assuming that the environment where the agent moves is stored in a computational structure, which is or can be transformed into an undirected graph G(V,E), where *V* is the set of vertices containing its Cartesian coordinates and *E* is the set of edges connecting the vertices, with an associated non-negative cost from moving between them. Occupied areas are connected to the graph with a high flag value for their edges, so that it is realistic that they won’t be used as paths for search algorithms. Unknown areas may be labeled in the graph as free or occupied spaces. As the robot moves and gathers new information the graph is incrementally updated. One of the most common structures for world representation is the probabilistic occupancy grid [19], that discretizes the world in a grid with cells, where each cell represents a portion of the world, and has an associated binary value, occupied or free. We may consider each center of the grid cells as a vertex, and adjacent cells are connected through edges.

Commonly, in a graph representation of an occupancy grid, orthogonal edges are given cost *L* and diagonal edges $L\sqrt{2}$, where *L* is the cell size. In contrast, a different approach is found in [20]. The Theta* algorithm is an “any-angle” path planning algorithm whose paths are not constrained by orthogonal and diagonal edges of a grid. In our case, we apply the approach described in [19]. Figure 1 shows a graph representation of an occupancy grid, where obstacle cells were removed from the graph.

For the specific path problem dealt with in this paper, we imposed some constraints that approach real world situations. They are:(i)The agent does not have access to any world information a priori.(ii)The environment map is built incrementally during navigation, based only on on-board sensors, and the agent will not perform special maneuvers (such as, intentional stops or rotate on its own axis) to gather more information.(iii)The agent will be provided with destination goals (points in the Cartesian world), and the global path planning algorithm has no time-processing constraints to find a path (if there is any).(iv)Once the agent starts moving, the motion controller must provide control signals every 100 ms, which will be used as the time-processing constraint for the local path planning algorithm.

Due to the problem constraints, a local path planning algorithm as solution for obstacle avoidance of unknown objects is performed, instead of searching for a new optimal path. The global path-planning, which is executed before starting the navigation, is done using classic A* algorithm. Aiming to get a faster response for unpredicted events, the tangential escape algorithm is run to avoid obstacle collisions. Although this approach is not optimal in terms of path solution, it has an O(1) complexity for any map given, not requiring a high computation effort.

Heuristic methods such as genetic algorithms (GA) [21], neural-networks (NN) [22], ant colony optimization (ACO) [23] and particle swarm optimization (PSO) [24] usually show good results in terms of path planning for uncertain environments, but they can be very computationally expensive to converge to an optimal solution [25]. On the other hand, sampling based methods, like the classical Rapidly-Exploring Random Trees (RRT) and Probabilistic RoadMap (PRM), can overperform heuristic methods in real world uncertain situations [26], but may require more accurate sensors, thus more expensive on-board hardware [25]. Therefore, these requirements are contrary to the proposal of this work, which aims at the path-planning solution using sensors and processors with limited capacities.

## 3. The Hybrid Path-Planning Strategy

The general idea of the hybrid planning is that, a destination goal is given, and before the start of the movement, it is searched for a complete and optimal path in the mapped area, using A* search. If none is found, the agent will move reactively through the environment, using only the local planning strategy, the tangential escape, to avoid collisions with obstacles. In practice, it is similar to trace a straight line from the depart point to the goal one, and avoid obstacles between them. If a path was obtained, the robot will follow it in order to avoid obstacles within the mapped environment and will again use the tangential escape to overcome unknown obstacles. This strategy is adopted to maintain a low computational cost of the algorithm, considering that the solution does not search for a new complete path in face of a unpredicted obstacle. Instead, the local planner is used to avoid the collision and the robot returns to the previously planned path. Also, the mapping of the environment occurs constantly using sensor data to incrementally update the occupancy grid of the environment.

The key point of our proposal is the integration between the global and the local path-planning. In other words, the robot navigates reactively, whenever it does not have enough information about the environment, or deliberatively when an A* path is provided. Nevertheless, there are some case where the robot should abandon the A* path to avoid a collision, and after overcoming the obstacle, it should recover the route in the closest point.

The A* Algorithm [1] performs a heuristic search in the map (modeled as occupancy grid). The algorithm keeps two lists, an open one which stores expanded nodes yet to be evaluated, and a closed one, which stores nodes already evaluated. The algorithm always searches through the open list for the node *n* which minimizes the function
f(n)=g(n)+h(n),where g(n) is the cost of the current path from the origin to the node and h(n) is the heuristic function that estimates the cost from *n* to the goal node.

The search will go on until a path is found or a complete path does not exist. Performing a single A* search is relatively fast for small areas, even though high resolution (portion of the real world that each cell represents) grids may leave the algorithm slower. The search strategy will look for a path within the mapped area, thus the unknown zones will be considered obstacles. After having the entire environment mapped, it is expected that the A* finds global optimal paths. Figure 2 shows an example of A* for an occupancy grid.

Assuming that distance sensors are being used for obstacle detection, the tangential escape [11] sets up a limit $d_{obs}$ for the robot to avoid obstacles. If something is detected in a distance less or equal to $d_{obs},$ the algorithm calculates a temporary goal to which the robot will move tangent to the obstacle, as illustrated in Figure 3. To achieve such maneuver, a rotation angle γ is computed, to modify temporally the current destination goal.

The angle between the minimum distance to the obstacle and the robot’s current orientation, β, and the angle between the linear velocity vector and the distance vector to the goal α, are used to calculate γ, which is given by
(1)$$\gamma =\left\{\begin{array}{cc} -\frac{\pi }{2}-\alpha +\beta , & \mathrm{if}\;\beta <0\\ +\frac{\pi }{2}-\alpha +\beta , & \mathrm{otherwise} \end{array}\right.$$

The main advantage of this method is its simplicity and efficiency, fitting to our problem’s constraints, even though its optimal performance cannot be guaranteed. Another drawback is the case of local minima, but the method is still very confident for obstacle avoidance for simple obstacles. Figure 3 shows the tangential escape strategy. Details of tangential escape strategy for more complex obstacles can be found in [11].

The switches between global and local strategies will occur based on the sensor measurements and $d_{obs}$ limit. If the agent is following a A* path, it must abandon it whenever any measurement is equal or less than $d_{obs}$. Once the obstacle is no longer in the limit, the robot should recover to the previous path. Knowing that the robot deviations takes it away from the path, a new returning point must be calculated to complete the avoidance task. A mechanism to prevent the robot from continuing deviating (circulating) the obstacles as shown in Figure 4a is also needed. To address this issue, a forgetting factor (fe∈[0,1]) retains a fraction of the last computed γ, so that the robot has a smooth rotation before tracking a new target. Then, the new angle is now obtained by
(2)$$\gamma \left[k\right]=\left(1-fe\right)\gamma \left[k-1\right]+\gamma_{c}\left[k\right]fe,$$where $\gamma_{c}$ is the current γ value. Notice that it is assumed that γ[k−1] value is known (with initial value 0). The pseudo-code for the tangential escape is shown in Algorithm 1.

**Algorithm 1:** Tangential Escape Algorithm.**Require:** Input: γ, fe, $d_{min}$, $d_{obs}$
**Ensure:** Output: Virtual Target Position $X_{v}$
1:$X_{v}\leftarrow X_{d}$2:**if**$d_{min}<d_{obs}$**then**3: β←ClosestSensorMeasurementAngle4: $\theta \leftarrow \arctan(y_{g}/x_{g})$5: α←θ−ψ6: **if**
β<0
**then**7:  γ←−π/2−α+β8: **else**9:  γ←+π/2−α+β10: **end if**11: $d\leftarrow norm(X_{v}-X)$12: $X_{v}\leftarrow X+\tanh\left(d\right){\left[\begin{array}{cc} \cos(\theta -\gamma ) & \sin(\theta -\gamma ) \end{array}\right]}^{T}$13:**else**14: γ←γ(1−fe)15:**end if**16:**return**$X_{v}$


The returning point, after avoiding the obstacle, is the closest one from the A* path to the robot, not considering any previous points already reached. Figure 4b illustrates this solution.

It is expected that the A* finds optimal collision free paths (apart from unknown or moving obstacles), so $d_{obs}$ limit can be reduced during the path-following mission, and it is now called $d_{obsA}$. This new limit has to be lower than the minimum distance from the path to any mapped obstacle (safety zone), otherwise the robot will deviate with no need. The pseudo-code code for the hybrid planning is given in Algorithm 2.

**Algorithm 2:** Hybrid navigation algorithm.**Require:** Occupancy Grid OG, Robot current Pose *X*, Robot Desired Pose $X_{d}$, γ, fe
**Ensure:** Occupancy Grid OG, $X_{d}$
1:$route\leftarrow AStarSearch(OG,X,X_{d})$2:**if**routeWasFound**then**3: $path\leftarrow GeneratePath(OG,X,X_{d})$4: PathFound←true
5: $d_{obs}\leftarrow d_{obsA}$
6: Aescape←false
7:**else**8: $d_{obs}\leftarrow d_{obsT}$
9:**end if**10:**while**$X\neq X_{d}$**do**11: X←GetRobotOdometry()
12: M←GetSensorData()
13: UpdateMap(OG,M,X)
14: **if**
PathFound
**then**15:  $\begin{array}{c} \mathrm{if}\;(min\left(M\right)\leq d_{obsA}\;\wedge \neg \left(Aescape\right))\;\vee \\ \left(Aescape\wedge min\left(M\right)\leq d_{obsT}\right) \end{array}$
**then**16:   $d_{obs}\leftarrow d_{obsT}$
17:   Aescape←true
18:  **else**19:   $d_{obs}\leftarrow d_{obsA}$
20:   Aescape←false
21:  **end if**22:  Controller←PathFollowingController
23:  $X_{d}\leftarrow getClosestPathPpoint$
24: **end if**25: $X_{d}\leftarrow TangentialEscape\left(min\left(M\right),d_{obs},fe,\gamma \right)$
26: $U\leftarrow Controller(X,X_{d})$
27: sendControlSignals
28:**end while**29:**return**OG,X


## 4. Results and Discussion

The experiments are run using Pioneer-P3DX, one of the most popular differential drive robots, in research labs. In its basic version, it weighs only 9 kg (with one battery) and can carry up to 23 kg. The robot comes with an on-board sonar sensor capable of providing distance measurements in eight different angles: One on each side, and six facing outward at 20-degree intervals. In this project, the resolution lack from sonar measurements is enhanced by the Kinect sensor, which is an RGB-D (RedGreenBlue-Depth) sensor with 60 degrees of field of view (FoV). In addition, the robot has a flat surface deck used for mounting the sensor and a personal computer (responsible for acquiring the sensor data and controlling the robot).

Initially the algorithm was tested in the mobile robot simulation environment MobileSim. The goal was to validate the navigation using the proposed algorithm and to evaluate the impact of two distinct heuristics, Euclidean distance and Manhattan distance within the paths obtained in the A* search. For the real world experiments, due the lower number of sonars of the P3-DX and the noise on their measurements, they are not suited for mapping. A solution was to integrate the sonar with a Xbox 360 Kinect, during the mapping and the obstacle avoidance.

The kinematic position controller used in this work is detailed in [11], as well the strategy to leave the virtual target and to recover the desired one. This work is recommended as reading for a stability analysis of the controller and its performance at different scenarios.

The parameters chosen to evaluate the efficiency of the navigation are the traveled distance, time of navigation, position error, which is the distance between the robot current position and the destination point, and the IACS (integral of absolute control signals), an estimate of the energy spent through the executed control signals and calculated by $\int_{0}^{t}∥u∥dt$, where *u* is the velocity developed by the robot.

For the mapping problem, we used the occupancy grid obtained through ROS (Robot Operating System) wrapper of RTAB-Map (real-time appearance-based mapping), a RGB-D SLAM (Simultaneous Localization and Mapping) very popular and with great performance using the kinect RGB-D camera. The RTAB-Map node is responsible for providing both the occupancy grid and the odometry corrections [27]. In order to fit in realistic applications, external localization solutions (ground truth), such as cameras or landmarks were not used.

### 4.1. Simulations

The simulation environment is shown in Figure 5a. Given the fact that the simulator is not susceptible to measurement and odometry noises, the mobile robot used for navigation was the standard Pioneer P3-DX with its eight sonar sensors, as mentioned before. The robot goal was to move between points A and B, starting at A, in a A-B-A-B circuit. Each trajectory from now on will be identified as follows: S1, for the first A-B, S2 for the first B-A, and S3 for the second A-B. For each trajectory, the respective illustration Figure shows the map obtained at the end of it. As value for the observable distance was used $d_{obsT}=$1 m and $d_{obsA}=0.4$ m.

Initially at S1 (Figure 5b), because there is not prior information and unknown areas are considered obstacles, the robot navigates reactively executing the tangential escape. It can be noticed by the trajectory followed that the robot is incapable of detecting the obstacle at all times during the deviation, causing oscillations in the obstacle avoidance path. Another observation is the path taken for the deviation at the uppermost obstacle, which is the least visible one near the goal. This can be explained by the limited area covered by the sonar and the reactively aspect of the tangential escape.

At S2 (Figure 5c), the robot possesses the mapping from the previous trajectory. Thus, it was possible to obtain a path with the A* search. Four cells, where each represents 0.1 m × 0.1 m of the environment, were used as safety zone. It can be noticed that the obtained path was enough to provide the obstacle avoidance, without the need to execute the Tangential Escape.

At S3, again a path was obtained in the A* search, but this time a different from the previous (Figure 5d). This happens because more information about free zones and obstacles was added to the map, modifying the occupancy grid. When the robot reaches the area highlighted by a rectangle, an obstacle is detected closer than the $d_{obsA}$ defined, causing the algorithm to perform a tangential escape. After the obstacle avoidance, the robot returns to the path initially planned and continues until it reaches the goal. The results from the navigation are shown in the Table 1. The video of this simulation can be seen https://www.youtube.com/watch?v=AihmDUMxU9Q.

To evaluate the efficiency of the two most common heuristics for the A* search, the Euclidean distance and Manhattan distance, another simulation was set within the same map of the first simulation. This time, the robot objective was to perform the A-B and B-A trajectories 21 times, with 42 movements total. This time it was used a 0.05 m × 0.05 m resolution for the occupancy grid, using the same measurements of the previous simulation for the evaluation of the navigation efficiency. The results are shown in Table 2.

### 4.2. Real World Experiments

The real world experiments were run in a room of approximately 50 m${}^{2}$ (Figure 6a). Rectangular boxes were used as obstacles. In the first one, the objective was to move the robot from A(x,y = [0 m, 0 m]) to B(x,y = [6.25 m, −2.4 m]), then return to A, go again to B and finally return to A, completing the cycle A-B-A-B-A. From now on, the trajectories will be named E1, E2, E3, and E4 following the sequence described. The obstacles were set in way that the tangential escape would lead the robot in different ways, depending on the control signals and odometry noises. Only two desired points was defined mainly to demonstrate how the A* path works for a map that was not completely discovered. It is important to emphasize that if the robot moved randomly in the room, it eventually would get it all mapped and the best A* path will be found. However wandering missions are not performed in this work, in contrast the robot knows the goals it should reach, even if it is a temporary one.

After having a portion of the mapped area, the A* search was performed and the path with the lowest risk of collision was obtained. Considering that each cell in the occupancy grid represents 0.1 m × 0.1 m in the real world, and due to the large physical dimensions of the Pioneer P3-DX, four adjacent cells was used to define the safety zone. For the tangential escape strategy, the variables were $d_{obsT}=0.85$ m and $d_{obsA}=0.4$ m.

Figure 6 shows the mapped area at the end of each movement. First, E1 (Figure 6b), the robot navigates reactively with the simultaneous mapping of the environment. At E2 (Figure 6c), the mapping was shown to be insufficient for obtaining a path with the A* search. This happened because of the defined safety zone, which caused the occupancy grid to be disconnected with any free cell path from the robot position to the goal (Figure 7a,b). At E3, the previous navigation was shown to be sufficient enough to obtain an A* path. In addition, the correction of odometry through RTAB-Map’s SLAM algorithm, using the loop closure detection can be noticed. This affects not only the robot position but the current map, a fact that can be noticed comparing Figure 6c,d. Finally at E4 (Figure 6e), a path like the one obtained at E3 is provided. This time, more noises in the odometry and sensor measurements causes the final map to be less accurate than the previous, since no loop closure was detected in the trajectory. The Table 3 shows the results of the navigation. The video for this is experiment can be seen https://www.youtube.com/watch?v=1mzWuNmF2Sg.

The second experiment analyzes the switch between global and local algorithms. The environment is shown in Figure 8. Before moving the robot receives the entire room’s map as a collision-free zone. No obstacles were previously mapped, so the obstacles in front of the robot are not represented in the map. Thus, the robot will navigate in a hybrid way, i.e., deliberatively during the path following, and reactively if an obstacle is detected in the route. In the last case, after overcoming the obstacle, the robot should recover the path at the closest point.

In this experiment the robot should move 5 m straight ahead (*x* = 5 m, *y* = 0 m), while avoiding an I- and V-shape obstacles, as can be seen https://www.youtube.com/watch?v=LJaxxm4_sII.

For both situations, four different parameters for $d_{obsA}$ and $d_{obsT}$ were tested. Figure 8 shows the robot’s path through each situation. It can be noticed that the low sensory capabilities causes the robot to oscillate, while it is performing obstacle avoidance. Whenever it “loses” sight of the obstacle it tries to go back planned path. Such behavior also increases the robot’s odometry error. As a consequence it can “think” the goal is already reached, but actually it is still a considerable distance from the desired destination. Finally, the experiments also showed that when facing an obstacle completely orthogonal with the planned path, the robot does not complete the avoidance task for very low $d_{obs}$ values ($d_{obsA}$ = 0.55 m and $d_{obsT}$ = 0.85 m, for instance).

In this work, the robot’s 0.4×0.4 m${}^{2}$ quadratic dimension is equivalent to at least 24×24 cells in the discretized environment, i.e., 1.2×1.2 m${}^{2}$ in the real map, when its safety regions are considered. Thus, to conduct the experiments in the available space, it was not possible to arrange obstacles too close to each other, otherwise passage routes would be considered closed ones. Nonetheless, we understand that it implies in facilitating some obstacle avoidance maneuvers, but it does not invalidate the proposal, because the focus of this work is hybrid navigation in multiple back-and-forth tasks.

## 5. Conclusions

This paper introduced a hybrid algorithm for the path planning problem. The strategy was conceived for low cost platforms at real life situations, where there are limited sensor abilities, no prior information of the environment and odometry noises. The combination of two simple, yet effective, local and global planning makes the robot capable of handling most situations, even though the optimal path cannot be guaranteed. The use of eight sonar sensors for obstacle avoidance, although it has a good deal of noise and covers limited area, was enough to avoid collisions with a relatively large safety zone. Furthermore, these limitations make it unfeasible to map using only the sonars, so the Kinect sensor was used to aid the mapping task.

Despite the success in the numerical validations, the sonar sensors did not demonstrate satisfactory results in real experiments, due its unreliable and imprecise measurements, over very great or very close distances. Moreover, the odometry sensor seems noisy, affecting the mapping and the robot self-localization. Both situations are effectively enhanced by the RTAB-Map toolkit integrated with the Kinect sensor. It is worthy mentioning that the navigation and mapping is fulfilled still with a low-cost platform, in terms of hardware and software.

During navigation, environment mapping results in incomplete maps presenting none or only partial obstacle location data. Considering this situation in addition to the robot’s physical dimensions, safety zones present themselves as essential for applying an A* search. Experimentally, the algorithm can find a feasible collision-free path in the mapped area, even when safety zones are adopted. In addition, our results demonstrate that the robot can leave the planned path to avoid an unpredicted obstacle, and then recover it by using hybrid control. Notice it occurs because the reactive algorithm activated to deal with the new obstacle.

The navigation strategy proposed can be applied in various service robot situations, particularly for aiding object displacement in work environments such as libraries, storage warehouses and mass production lines. These environments usually require repetitive going back-and-forth tasks on a predefined path on partially observable surroundings. In another perspective, home cleaning robots can lessen traveled distance by improving navigation, turning more effective and energy-efficient as the environment becomes well-known with time.

For future works, different situations and the usage of sensors capable of covering a larger area (at least for obstacle avoidance task) may be experimentally tested, to evaluate the effectiveness of the algorithm in more cluttered and dynamic environments.

## Figures and Tables

**Figure 1 sensors-19-01049-f001:**
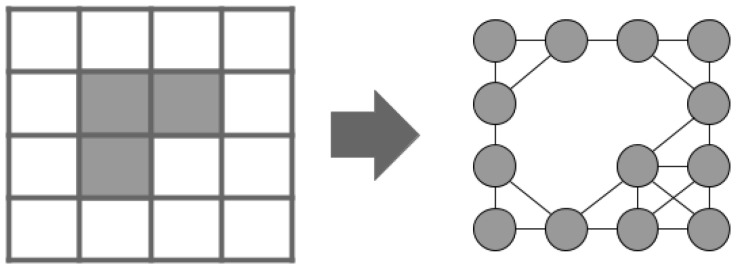
Undirected graph representation of a 2D occupancy grid.

**Figure 2 sensors-19-01049-f002:**
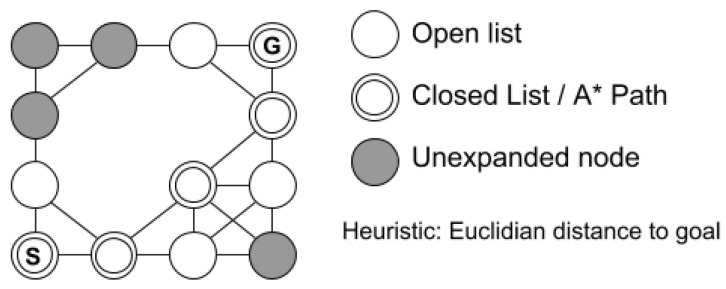
A* search.

**Figure 3 sensors-19-01049-f003:**
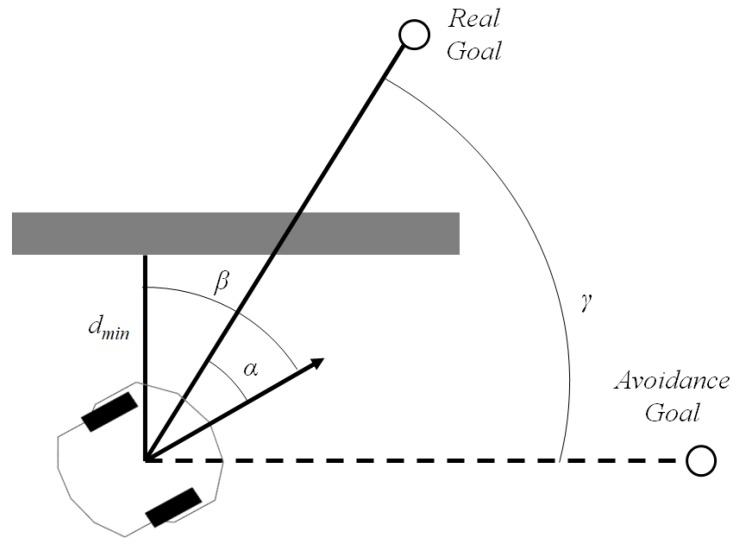
Tangential escape strategy.

**Figure 4 sensors-19-01049-f004:**
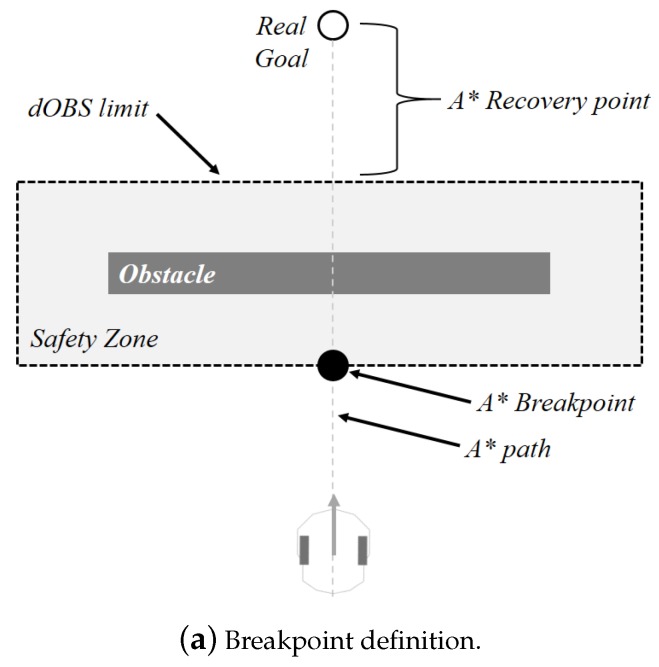
Tangential escape and recovery point definition.

**Figure 5 sensors-19-01049-f005:**
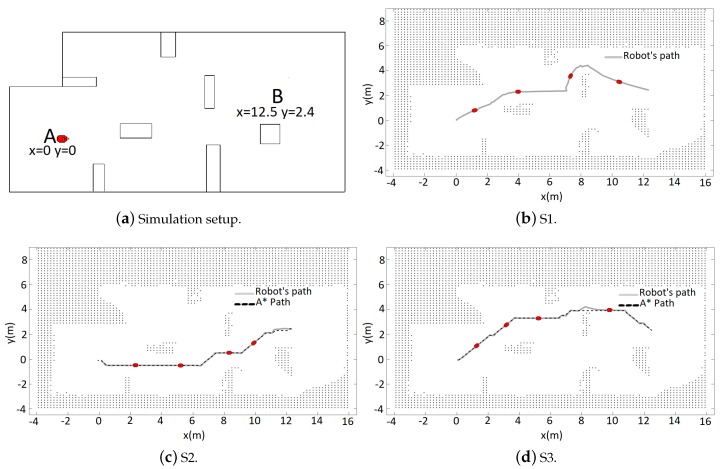
Navigation and mapping.

**Figure 6 sensors-19-01049-f006:**
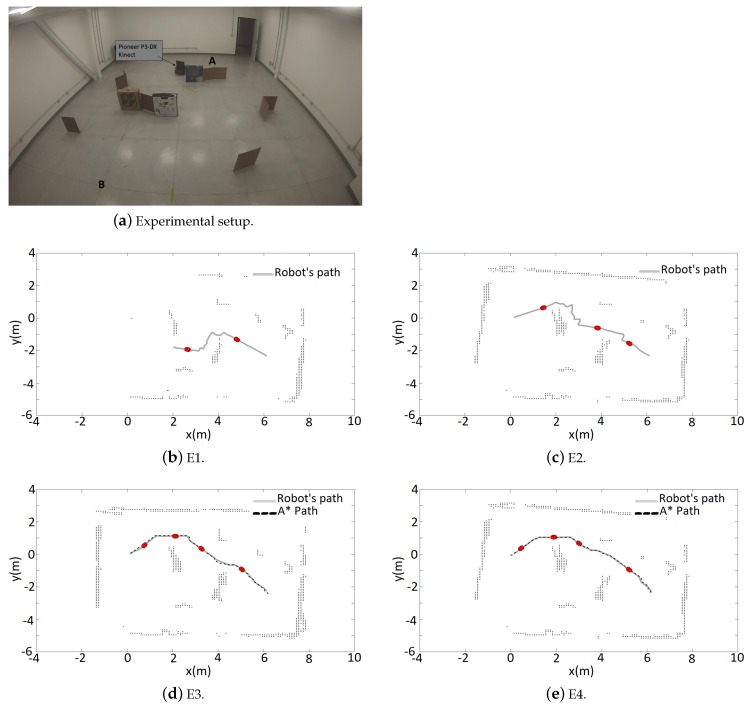
Navigation and mapping.

**Figure 7 sensors-19-01049-f007:**
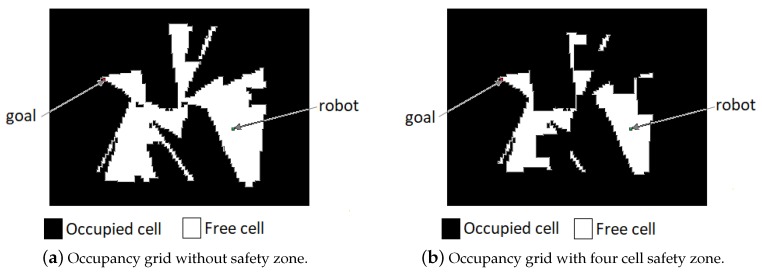
Comparison of occupancy grid without/with safety zone.

**Figure 8 sensors-19-01049-f008:**
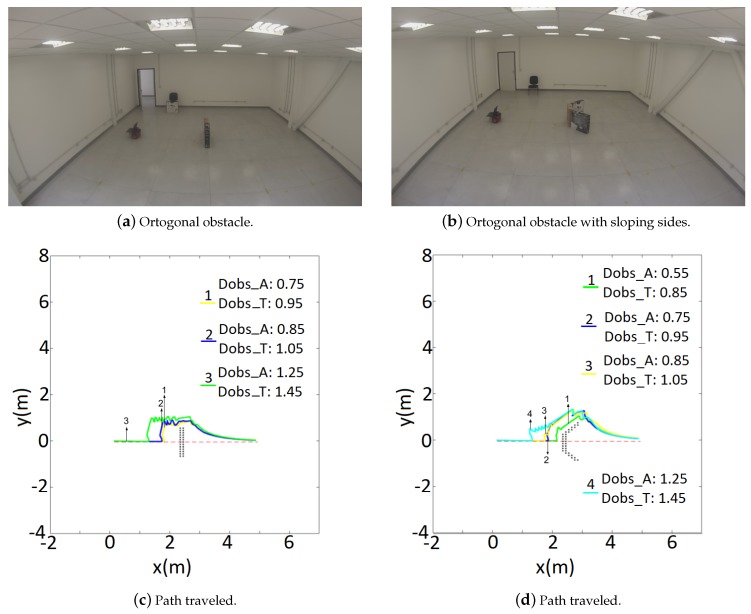
Comparison of tangential escape parameters for a straight line experiment.

**Table 1 sensors-19-01049-t001:** Results obtained from the simulation. Integral of absolute control signals (IACS).

	Trajectories		
	**S1**	**S2**	**S3**
Distance (m)	15.22	13.82	14.99
Linear Velocity (m/s)	0.13 ± 0.06	0.18 ± 0.05	0.17 ± 0.04
IACS	0.15	0.15	0.17
Time (s)	90.55	59.18	67.88

**Table 2 sensors-19-01049-t002:** Heuristic comparison.

	Euclidian	Manhattan
Total distance (km)	0.62	0.62
Average distance each movement (m)	14.88	14.83
Minimum distance for one movement (m)	13.37	13.33
IACS	9.76	9.14

**Table 3 sensors-19-01049-t003:** Real world experiment.

Data/Trajectory	E1	E2	E3	E4
Distance (m)	8.52	8.53	7.97	7.97
Linear velocity (m/s)	0.10 ± 0.06	0.10 ± 0.06	0.18 ± 0.03	0.18 ± 0.03
IACS	1.00	0.48	0.22	0.28
Time (s)	74.86	75.88	42.48	41.96

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
