# Peer review of "A Hybrid Path-Planning Strategy for Mobile Robots with Limited Sensor Capabilities"

_sensors, 2019, doi:10.3390/s19051049_

Reviewer 1 Report

This paper introduces a strategy for the path planning problem for platforms with limited
2 sensor and processing capabilities. Topic is actual and interesting.

Introduction is in suitable form.

Coments:

- Figure 6 has small text no axes.

- It could be good to write short description of used robot properties and sensors (robot geometry, sensor placement, control unit, etc.)

-  Conclusion is very short.

Author Response

Dear Reviewer #1,

Thank you very much for your comments. We really appreciate your collaboration and hope to have healed your doubts, completely contemplated your suggestions and answered your questions.

Our modifications have been ordered in this note accordingly your review and marked in blue color over the revised version of the paper.

Sincerely yours,

Alexandre Santos Brandão

Reviewer Comments and our answers

This paper introduces a strategy for the path planning problem for platforms with limited 2 sensor and processing capabilities. Topic is actual and interesting.

Introduction is in suitable form.

Thank you very much.  We understand that robotics is an important area where we must propose, whenever possible, simple and efficient solutions using limited resources for daily problems.  

Coments:

- Figure 6 has small text no axes.

We replaced the old figures for new ones, with more readable numbers.  

- It could be good to write short description of used robot properties and sensors (robot geometry, sensor placement, control unit, etc.)

To attend your suggestion, we changed and included new comments in the first paragraph of Section 4.

The experiments are run using Pioneer-P3DX, one of the most popular differential drive robots, in research labs. In its basic version, it weighs only 9 kg (with one battery) and can carry up to 23 kg. The robot comes with on-board sonar sensor capable of providing distance measurements in eight different angles: one on each side, and six facing outward at 20-degree intervals. In this project, the resolution lack from sonar measurements is enhanced by the Kinect sensor, which is an RGB-D sensor with 60 degrees of Field of View (FOV). In addition, the robot has a flat surface deck used for mounting the sensor and a personal computer (responsible for acquiring the sensor data and controlling the robot).

-  Conclusion is very short.

To attend your suggestion, we complement the conclusion with the following paragraphs.

Despite the success in the numerical validations, the sonar sensors did not demonstrate satisfactory results in real experiments, due its unreliable and imprecise measurements, over great or very close distances. Moreover, the odometry sensor seems noisy, affecting the mapping and the robot self-localization. Both situations are effectively enhanced by Rtabmap toolkit integrated with the Kinect sensor. It is worthy mentioning that the navigation and mapping is fulfilled still with a low-cost platform, in terms of hardware and software.

During navigation, environment mapping results in incomplete maps presenting none or only partial obstacle location data. Considering this situation in addition to the robot’s physical dimensions, safety zones present themselves essential for applying A* search. Experimentally, the algorithm can find a feasible collision-free path in the mapped area, even when safety zones are adopted. In addition, our results demonstrate that the robot can leave the planned path to avoid an unpredicted obstacle, and then recover it by using hybrid control. Notice it occurs because the reactive algorithm activated to deal with the new obstacle.

 The navigation strategy proposed can be applied in various service robot situations, particularly for aiding object displacement in work environments such as libraries, storage warehouses and mass production lines. These environments usually require repetitive going back-and-forth tasks on a predefined path on partially observable surroundings. In another perspective, home cleaning robots can lessen travelled distance by improving navigation, turning more effective and energy-efficient as the environment becomes well-known with time.

Reviewer 2 Report

The paper under review shows a hybrid approach to the path planning problem that tries to integrate A* planning as the global approach and Tangential Escape as the local planner.

The method is interesting given its applicability for limited capabilities robotic platforms (both in sensor and processing capabilities). However, I consider that experimentation is too shallow and that to be ready for publication, more simulation and experimental work has to be done.

For simulation, I think a good benchmark could be to show that the method works in the maps from the  Repository Motion planning maps of the Intelligent and Mobile Robotics Group from the Department of Cybernetics, Czech Technical University in Prague.These maps are available at http://imr.felk.cvut.cz/planning/maps.xml. Several queries need to be run on each map to show the performance of the proposed method.

Real experimentation could include more complex setups, even if it is known that is difficult to configure in small working areas.

The paper is well-written and only small typos are in. An example is the heading of the Tangential Escape algorithm description, where it is written (Tangencial).

Author Response

Dear Reviewer #2,

Thank you very much for your comments. We really appreciate your collaboration to improve our work. This enhances us scientifically.

Specifically speaking, we run several numerical simulations in order to attend your suggestions, however in the suggested scenarios it was no possible to evaluate our proposal, once open passages are dealt as closed ones due the required safety zone (obstacle dilatation) during A* path-searching algorithm. We address all explanation following and we hope to clarify possible doubts.

Our modifications have been ordered in this note accordingly your review and marked in green color over the revised version of the paper.

Sincerely yours,

Alexandre Santos Brandão

Reviewer Comments and our answers

The paper under review shows a hybrid approach to the path planning problem that tries to integrate A* planning as the global approach and Tangential Escape as the local planner.

The method is interesting given its applicability for limited capabilities robotic platforms (both in sensor and processing capabilities). However, I consider that experimentation is too shallow and that to be ready for publication, more simulation and experimental work has to be done.

For simulation, I think a good benchmark could be to show that the method works in the maps from the Repository Motion planning maps of the Intelligent and Mobile Robotics Group from the Department of Cybernetics, Czech Technical University in Prague. These maps are available at http://imr.felk.cvut.cz/planning/maps.xml. Several queries need to be run on each map to show the performance of the proposed method.

Thank you very much for your remarks. I also agree more simulation and experiments could be done, however we understand the proposed algorithm is able to observe an environment, store features in a discretized map and then establish an optimal (whenever available), after performing some interactions of a going back-and-forth task.

We have checked the suggested maps (on http://imr.ciirc.cvut.cz/planning/maps.xml) since the previously informed link was broken. I hope you was talking about this one.

In our point of view, the suggested maps can be classified as:

a)      Traps: bugtraps, T-shape, slits easy, square spiral, unsolvable

b)      Narrow corridor, close obstacles or cluttered environments: brick_pattern, complex3, corridor, dense, geometry, large, maze, tunnel

c)       Single path solution: Clasps, back and forth, complex, rooms

d)      Similar to our setup: Complex2, gaps, warehouse.

We have conducted the required simulations, however we noticed most of them present a single feasible route or path in both reactive and deliberative planning strategies. In that sense, the maps are extremely useful as robot navigation traps, but nevertheless do not match our proposed goal, which is finding alternative routes in a back-and-forth task. As mentioned before in the manuscript, obstacle avoidance tasks have been explored in another article [11] and that our present focus is precisely on finding alternative routes after a few iterations of going to and coming back displacement. Furthermore, the strategy is able to decide whether it is necessary to reactively act upon noticing a previously unobserved obstacle or even if it is a new object on the map (dynamic observation condition). After overcoming the unpredicted obstacle, the strategy also defines the next closest return-point at the predefined route, which is the safest path established by the safety zone.

[11] Brandão, A.S.; Sarcinelli-Filho, M.; Carelli, R. An analytical approach to avoid obstacles in mobile robot navigation. International Journal of Advanced Robotic Systems. 2013, 10, 278.

It is worth mentioning that the algorithm is proposed to be collaboratively used with humans in our future works. In such a case, the robot will execute a leader-following task until a certain point defined by that leader (human). After finishing the accompanying mission, the robot must return to its departure position. In that stage, the robot will be able to optimize its way back in case it noticed an optimal route during its observations in the leader-following trajectory.

Real experimentation could include more complex setups, even if it is known that is difficult to configure in small working areas.

In the available experimenting space, the disposition of obstacles directly affected the strategy validation. After all, close obstacles, once dilated by the safety zone, make feasible free-navigation routes, closed ones, during the A* algorithm’s search for an optimal path.

That is also the reason why some maps suggested in the link became impractical to test our proposition, as regressing after going from a point A to B was unfeasible according to A*, because of the obstacle proximity, caused by dilation.

It is important to remark that in this present work; the map cells have been discretized into 0.05 x 0.05 m² dimension. Given the robot’s 0.4 x 0.4 m² quadratic dimension, for a collision-free navigation, we recommended and adopted safety regions of that same dimension. In other words, a collision-free route for the robot is equivalent to at least 24 x 24 cells in the discretized environment, i.e., 1.2 x 1.2 m² in the real map. However, to conduct the experiments, it was not possible to arrange obstacles too close to each other, otherwise passage routes would be considered closed ones. Nonetheless, we understand that it implies in facilitating some obstacle avoidance maneuvers, which does not invalidate the proposal, given that the focus of this work is hybrid navigation in multiple back-and-forth tasks.

The paper is well-written and only small typos are in. An example is the heading of the Tangential Escape algorithm description, where it is written (Tangencial).

Thanks for your observation, we check the typos and correct them.